# Mycotoxin Prevalence and Microbiological Characteristics of Locally Produced Elected Freekeh Products

**DOI:** 10.3390/toxins16110499

**Published:** 2024-11-20

**Authors:** Samer Mudalal

**Affiliations:** Department of Nutrition and Food Technology, Faculty of Agriculture and Veterinary Medicine, An-Najah National University, Nablus P.O. Box 707, Palestine; samer.mudalal@najah.edu

**Keywords:** freekeh, microbiological, physical impurities, mycotoxins

## Abstract

Freekeh is produced from roasted, immature wheat grains. It is very popular in Middle Eastern and North African nations. This study aimed to evaluate the occurrence of different types of mycotoxins, physical impurities, and microbiological contamination in local freekeh products. Lateral flow competitive immunochromatographic assay was used to evaluate the occurrence of mycotoxins. It was found that physical impurities for some tested products exceeded the permitted limit (>2% of straw and foreign grains). Moreover, our findings showed that total aerobic bacterial and fungal counts in Freekeh products varied from 1 to 4 logs and from 1.39 to 4.3 logs, respectively. The incidence ranges of aflatoxins and ochratoxin were 3.17–3.33 ppb and 4.63–8.17 ppb, respectively. The levels of deoxynivalenol (DON) and T2/HT2 (trichothecene T2 and deacetylated form HT2) were less than the limit of detection. More than 78% of Freekeh samples tested had aflatoxin and ochratoxin contents higher than the limit permitted by the European Commission (4 and 5 ppb). In conclusion, gaining knowledge about the quality, safety, and labeling of freekeh products can help increase their commercial potential. Further investigations are needed to evaluate the factors affecting contamination levels within the freekeh supply chain.

## 1. Introduction

Freekeh, or firik, is a type of derived whole wheat product, and it is characterized by a smoked flavor. It is made from immature green wheat grains that are usually obtained by early harvesting (75 to 89 on the Zadoks scale). Freekeh is a very popular and traditional food product in many regions in Middle Eastern and North African nations [1]. Freekeh is usually manufactured by either boiling immature green wheat grains at atmospheric pressure or roasting (scorching) the young spikes on a grill to burn off the awns and leafy debris [2]. 

The production of freekeh is still very traditional and small scale [3]. Freekeh is characterized as having good nutritional value and high digestibility [4]. Freekeh has a high content of fructo-oligosaccharides and other functional compounds. Moreover, freekeh contains a low level of cholesterol and a high content of iron, fiber, and lutein [5]. Wheat grains in the milky stage are characterized by high levels of contents from the fructan group, which enhances the utilization of calcium and iron [6]. 

Wheat accounts for approximately 47% of Palestine’s major crops. Most of the wheat-cultivated areas are concentrated in northern Palestine. Durum is the major type of wheat (>70%) cultivated in the West Bank [7]. Freekeh is one of the most common wheat-derived products after wheat flour.

The risks of microbial contamination and the occurrence of mycotoxins is very high due to the traditional methods of manufacturing of freekeh (under uncontrolled conditions of relative humidity, temperature, sunlight, and rate of evaporation during roasting). There are few published studies about freekeh products, and most of them focus on quality traits, cultivation conditions, and manufacturing practices [3,5]. There are no available studies about the occurrence of mycotoxins in freekeh products. A huge number of studies have been published about the occurrence of mycotoxins in wheat grains and wheat-derived products, but not freekeh products [2,8,9,10,11].

Moreover, the traditional manufacturing process used to produce freekeh increases the risk of contamination by physical impurities (such as stones, gravel, dust, soils, insects, straw, foreign grains, rodent/bird residues, etc.). Unfortunately, the quality of commercially available freekeh products in local markets remains poorly understood. Accordingly, this study aimed to evaluate the compliance of locally produced freekeh with local and international regulations regarding the presence of mycotoxins, microbiological traits, and physical characteristics.

## 2. Results

### 2.1. Compliance of Freekeh Products with Physical and Chemical Label Requirements 

The descriptive results related to the compliance of freekeh products with the physical and chemical requirements of the Palestinian Standard for Freekeh (PS 4154/2019) are shown in Table 1. These requirements are mainly classified as color, odor, physical field impurities, and chemical proximate composition. Overall, all freekeh products were compliant with the requirements of the Palestinian Standard for Freekeh (PS 4154/2019) except freekeh products from companies MN, SKC, SKF, and SNN. In particular, the freekeh product from company MN had a content of physical impurities (straw and foreign grains) exceeding 2%. According to the Palestinian Standard for Freekeh (PS 4154/2019), it should be less than 2% [12]. In the case of freekeh products from companies SKC, SKF, and SNN, the protein contents were found to be below the standard (>11%).

### 2.2. Occurrence of Mycotoxins

The occurrence of zearalenone (ppb) in freekeh products is shown in Figure 1. The results showed wide variations in the levels of zearalenone between freekeh products. These variations were observed within the products from the same company (this was clear in products from companies SHQ, SKC, and NKH) as well as between companies. According to the European Commission, the maximum permitted level of zearalenone in wheat grains is 100 ng g^−1^ [13]. Accordingly, all tested freekeh samples were in compliance with the limit permitted by the European Commission [13,14].

Figure 2 shows the occurrence of ochratoxin (ppb) in freekeh products. Similar to zearalenone, there were variations in the levels of ochratoxin both within products from the same company and between products from different companies. The permitted level of ochratoxin in wheat grains, according to the European Commission, is 5 ppb [14]. The non-compliance rate for ochratoxin was very high; 78% of tested freekeh samples had ochratoxin levels higher than the limit permitted by the European Commission. 

Figure 3 shows the occurrence of aflatoxins (ppb) in freekeh products. The permitted level of aflatoxins in wheat grains, according to the European Commission, is 4 ppb [14]. More than 78% of tested freekeh samples exhibited aflatoxin levels higher than the limit permitted by the European Commission. There are no studies about the occurrence of aflatoxins in freekeh that are locally produced and available in the Palestinian market. Therefore, the results of this study were compared with some local products other than freekeh or with other products derived from wheat (freekeh is a derived wheat product). 

Five different types of mycotoxins (Zearalenone, DON, T2HT2, aflatoxins, and ochratoxins) were evaluated in local freekeh products (Table 2). The level of zearalenone ranged from 27.9 to 60.83 ppb. Freekeh products from SNN company showed the lowest content of zearalenone, while products from SHQ company had the highest level of zearalenone compared to other products. Regarding DON, it was lower than the limit of detection (<0.3 ppb) in all tested freekeh products, and at the same time, lower than the permitted limit according to the European Commission (600 ppb milling products of cereal). The summation of T-2 and HT-2 was found to be lower than the limit of detection (LOD < 50 ppb). Maximum levels (sum) of T-2 and HT-2 were set as 100 ppb (wheat, rye, and other cereals). The level of aflatoxins in freekeh products varied from 3.17 to 13.33 ppb. The lowest and highest levels of aflatoxins were observed in freekeh products from companies KS and MN, respectively. The products from companies MN and LZ exceeded the permitted limit. There were significant differences in the content of ochratoxins between freekeh products, and the range was 4.63–8.17 ppb. Most freekeh products from different commercial sources contained more ochratoxins than the European permitted limit.

The possibility of correlations between the occurrence of different mycotoxins was evaluated (Table 3). There was a significant negative correlation between the cooccurrences of aflatoxins and ochratoxins. There were no significant correlations between zearalenone, ochratoxins, and aflatoxins. For the remaining mycotoxins, correlations were not computed because the variables were constant.

### 2.3. Microbiological Analysis

Total aerobic bacterial and fungal counts were evaluated for freekeh products (Table 4). The level of bacterial and fungal contamination varied between products. Products from NKH company had the highest bacterial count (4.02 log cfu), while products from the MN company had the lowest count (1 log). Regarding fungal count, there were also significant differences between products. Overall results indicated that all tested products were microbiologically acceptable based on the Palestinian Standard for Freekeh (PS 4154/2019) except for products from SKC company, which were slightly higher than the standards (4.3 logs vs. 4 logs).

## 3. Discussion

### 3.1. Physical and Chemical Label Requirements 

The accuracy of nutrition labeling is crucial in food choices and can affect the purchase decision [15]. Moreover, labels should contain any related information regarding the handling and preparation of products [16]. Clear and reliable information about the identity and content of products is very important for consumers, and it should be written on food labels. In general, the regulations and mandatory information on food labels vary between countries [17]. In our study, the label regulations mentioned in the Palestinian Standard for Freekeh (PS 4154/2019) were used to evaluate the accuracy of labeled information versus actual values. In agreement with our findings, Fabiansson [18] found a significant discrepancy between declared and measured nutritional values on food labels in Australia. Similarly, Huang et al. [19] revealed that the majority of prepacked foods contained nutrition label information that did not adhere to Chinese nutrition labeling standards.

### 3.2. Mycotoxins 

Due to the lack of studies about the occurrence of mycotoxins in freekeh products, the results of this study were compared to wheat grains and wheat-derived products. Freekeh is a type of wheat-derived product. Although this comparison may lack precision, it can offer perspective. Our findings showed that all tested samples contained zearalenone below the permitted European limits. Fusarium head blight is responsible for the production of zearalenone. Outbreaks of Fusarium head blight have been associated with an increase in relative humidity [20]. It was found that the occurrence of zearalenone in wheat samples collected from Brazil was 56% [8]. Moreover, zearalenone was found in 35% of Italian wheat samples tested [21,22]. The percentage of wheat samples contaminated with zearalenone was 9% in Romania [23], 69% in Croatia [20], and 47% in Poland [11]. 

More than 78% of our tested freekeh samples contained ochratoxin. Elaridi et al. [24] found that only one out of 50 samples of wheat-derived products contained ochratoxin above the limit permitted by the European Union. A study conducted in Algeria found that 69.2% of wheat grains and 92.8% of goods derived from wheat were contaminated with ochratoxins [25]. A five-year study carried out in Canada between 2009 and 2014 found that over 50% of goods based on wheat, namely wheat, oats, and milled products of other grains, included ochratoxins [26].

Regarding aflatoxins, several studies showed that they have been found in significant levels in many food commodities in the Palestinian market, in agreement with our findings. Barakat and Swaileh [27] found aflatoxin in Palestinian multifloral honey. Moreover, the level of aflatoxins exceeded the permitted limits in some nuts, rice, and wheat flour that were marketed in Palestine [28]. Mudalal [29] found that 69% of tested za’atar mix samples contained aflatoxins. Iqbal et al. [30] found that 24–36% of wheat-derived products such as spaghetti, noodles, macaroni, and lasagna were contaminated with AFs. 

Our results regarding the incidence of zearalenone (27.93–60.83 ppb) were lower than that documented in the literature. A previous study showed that the incidence level of zearalenone ranged from 51 to 1135 ppb in tested wheat samples in Romania [10]. The highest incidence of zearalenone in wheat samples was 1000 ppb observed by Alexa et al. [23]. Pleadin et al. [20] revealed that wheat samples collected from the Croatian market had 7–107 ppb of zearalenone.

Our study showed that the level of DON in freekeh was less than the limit of detection (<0.30 ppb). On the contrary, Moretti et al. [21] found that DON was the most common mycotoxin detected in wheat in Italy. Our results were in alignment with Bertuzzi et al. [22], who found a low level of DON occurrence in soft and durum wheat. Rainfall, high relative humidity, and flowering temperatures (15 °C to 28 °C) are permissive to FHB disease and DON contamination [31]. The European Commission suggests that the combined amount of T-2 and HT-2 toxins in unprocessed wheat should not exceed 100 µg kg^−1^ [13,14]. Several researchers showed that the incidence of T-2 and HT-2 toxins was low in soft and durum wheat in Europe [32,33]. Calori-Domingues et al. [8] found that the average incidence levels of zearalenone and DON were 82 and 1046 ppb in wheat samples collected from Brazil. Zebiri et al. [25] revealed that the level of ochratoxin A in wheat grains ranged from 0.21 to 27.31 µg/kg. The range of the incidence of ochratoxins in wheat samples in India was 1.36–21.17 ppb [34].

There was no association between the amounts of DON and ZEN toxins, even though they are generated by the same fungus. This result was in agreement with the result observed by Ji et al. [35]. On the other hand, it was found that the occurrence of zearalenone was less than DON in barley [36].

### 3.3. Microbiological Analysis 

The level of bacterial and fungal contaminations varied between freekeh products (from 1 to 4 logs). Wheat grains are usually exposed to microbial contaminations at different stages: in the field (such as soil, insects, water, and animal feces), during harvesting, during processing, and during handling and storage [37,38,39]. Wheat grains can be contaminated by different types of bacteria (Micrococcaceae, Pseudomonadaceae, Enterobacteriaceae, etc.) and fungi (Alternaria, Cladosporium, Fusarium, etc.) [40,41].

Our results were in agreement with previous studies conducted on wheat grains. It was found that fungal counts ranged from 2 to 5 log CFU/g and 5 log CFU/g of aerobic bacterial count in 58 tested wheat samples [41]. Manthey et al. [31] revealed that durum wheat samples collected from growers, farm bins, and elevators contained a fungal count in the range of 1.4–5 log CFU/g. In Australia, a study showed that mean counts of yeast and mold in wheat samples were 3.7 and 2.7 log CFU/g, respectively [42]. A similar study in Hungary showed that mean counts of yeast and mold in organic wheat samples were 3.9 and 3.5 log CFU/g, respectively [43].

## 4. Conclusions

In conclusion, freekeh products that are available in the local market were not similar in quality attributes. The level of contamination by different types of mycotoxins was relatively high, and this was very clear in the incidence levels of ochratoxins and aflatoxins. Bacterial and fungal contamination also varied between products. This variation may be attributed to variations in the sources and the intensity of contaminations in the field (such as soil, insects, water, and animal feces), during harvesting, during processing, and during handling and storage. The levels of contamination by microorganisms and mycotoxins in freekeh products were relatively lower than those documented for wheat grains. Unique preparation operations such as flame roasting may contribute to this reduction. Further investigations are necessary to understand the role of roasting by flame in reducing the mycotoxin levels as well as the initial microbial load. Moreover, there is a need to improve the preparation process of freekeh to reduce its levels of physical impurities. 

## 5. Materials and Methods

### 5.1. Sample Collection and Label Information

Freekeh products were sampled from nine different local companies that were selling their products commercially in the market. From each company, three samples representing three different batches were collected, and from each batch two samples were collected. The total number of samples was 54 (9 companies × 3 batches × 2 samples/batch). Six samples from each company were selected for different quality attributes. The products of the companies were coded (AQ, LZ, KS, MN, NKH, SHQ, SKC, SKF, and SNN), and these codes were used to present the results. The products of the companies were selected based on their market shares. The nine companies were the leaders in the freekeh market.

Moreover, the label information (nutritional information, weight, type, and shelf life) of the freekeh products was recorded. Each packaging unit weighed about 900–1000 g. The samples were kept at room temperature under dry conditions for further analysis. After proper labelling, all samples were analyzed the day after arrival. Accordingly, there was no impact of storage conditions on the levels of the quality traits tested.

### 5.2. Proximate Composition

Freekeh samples were pulverized in a rotating mill made of stainless steel. In order to avoid the samples’ temperature rising while they were being ground, they were carefully and progressively added to the mill. To prevent contamination between samples, the grinding machine was carefully cleaned (with a brush) after every sample. Afterwards, the ground freekeh samples were tagged, placed in a dry and clean location in preparation for future testing, and sealed into plastic bags. Part of each sample was dedicated to determining protein, moisture, and ash contents. Moisture content was determined by measuring weight loss after drying using an air oven (Blender, Germany) at 105 °C for 16 h. Ash content was estimated with incineration at 525 °C for 4 h in a muffle furnace (Furnace Carbolite SN. 80 16919). Protein content was measured using the Kjeldahl method. All procedures related to proximate analysis were based on AOAC [44].

### 5.3. Determination of the Content of Physical Impurities 

All physical impurities, such as gravel, stones, straw, insects, soils, and foreign grains, were manually separated by spreading a 10 g freekeh sample on a white plastic plate (Figure 4). The removal, definition, and classification of physical impurities were carried out according to the Palestinian Standard for Freekeh (PS 4154/2019) which was approved and published by the Palestinian Standards Institute (PSI) [12]. The composition of physical impurities was divided into three classes: 

Stones, sand, and dirt

Dead insects and/or their parts

Other grains and stems.

**Figure 4 toxins-16-00499-f004:**
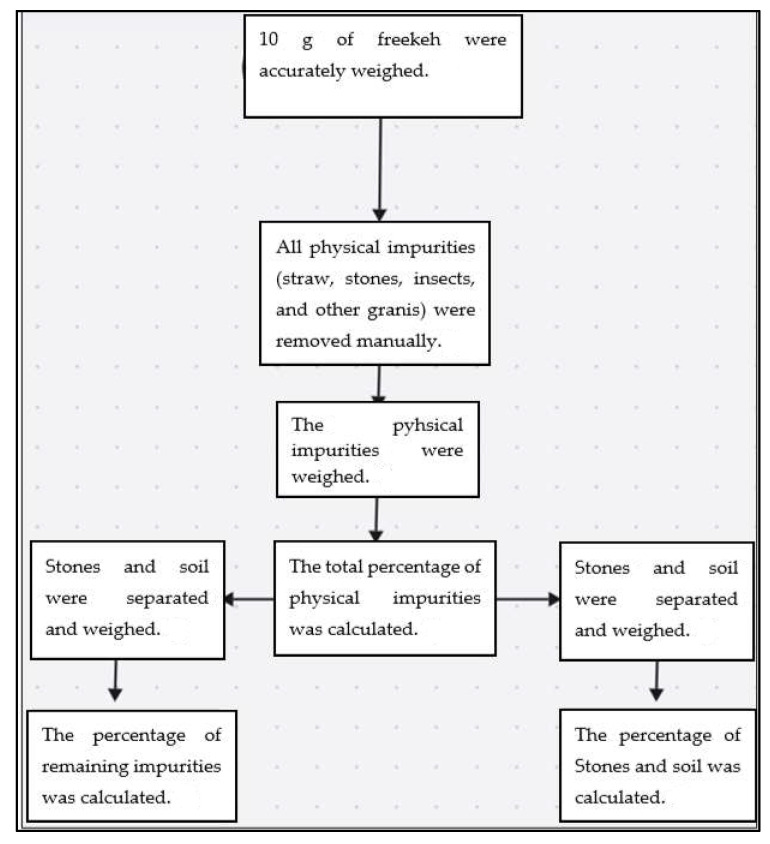
Flowchart procedure for the determination of physical impurities (stones, soils, insects, other grains, etc.).

### 5.4. Microbiological Analysis 

#### 5.4.1. Total Aerobic Bacterial Counts

From each company, representative samples were selected for microbiological analysis. A 10 g sample of freekeh was added to 90 mL of sterilized peptone in a stomacher pack. The mixture was homogenized by a stomacher machine for 1 min. Three dilutions (10^−1^ to 10^−3^) were used to count aerobic bacteria. Plate count agar (PCA) was used as a microbiological growth medium to estimate the total viable bacterial count. Each dilution was cultured in PCA in triplicates using Petri dish plates. The plates were incubated for 48–72 h at 37 °C. Plates containing 25–250 colonies were included for counting [45]. 

#### 5.4.2. Yeast and Mold 

From each company, representative samples were selected to determine fungal counts (yeast and mold counts). Potato dextrose agar (PDA) was used as a culture medium. Serial dilutions (10^−1^ to 10^−3^) were used to obtain proper colony counts. The plates were incubated for 4–5 days at room temperature. Plates containing 25–250 colonies were included for counting [45].

### 5.5. Determination of Mycotoxins

The method used in this study was a lateral flow competitive immunochromatographic assay to evaluate the incidences of different types of mycotoxins (Aflatoxins, DON, zearalenone, ochratoxins, and T2/HT2). The commercial kits (Reveal RQ+) for each type of tested mycotoxin, as well as the instrument used (AccuScan Gold reader) to read the results of tested samples, were purchased from Neogen Corporation (Ayr, Scotland, UK). 

The quantity of mycotoxins was determined according to the manufacturer’s instructions. Regarding aflatoxins, 10 g of each ground sample was mixed with 50 mL of 65% ethanol. For ochratoxin A, the same weight of each freekeh sample was mixed with 40 mL of 65% methanol. Both solvents were obtained from Sigma Aldrich (Munich, Germany). The obtained solutions were shaken vigorously for 3 min. The obtained solutions were left to settle for a few minutes. After that, 1 mL of solution was transferred to an Eppendorf tube and centrifuged at 2000 rpm for 1 min. One hundred μL of the sample extract (supernatant) was mixed with 500 μL of the sample diluent (from the Neogen kit) for aflatoxin analysis and 200 μL for ochratoxin A. One hundred μL of the diluted sample extract was transferred to the sample cup. For DON, 10 of the representative samples were mixed with 100 mL of distilled water. The mixture was shaken for 3 min and left to settle for 1 min. One mL of the supernatant solution was transferred to an Eppendorf tube and centrifuged at 2000 rpm for 1 min. One hundred μL of the sample extract was mixed with 1000 µL of the sample diluent in a red dilution cup. Regarding zearalenone, 10 of the representative samples were mixed with one MAX 1 packet. The mixture was added to 50 mL of distilled water and shaken for 3 min and left to settle for 1 min. One mL of the supernatant solution was transferred to an Eppendorf tube and centrifuged at 2000 rpm for 1 min. One hundred μL of the sample extract was mixed with 100 µL of the sample diluent in a red dilution cup. One hundred µL of the diluted sample extract was transferred to the sample cup.

With respect to T2/HT2, 10 of the representative samples were mixed with one MAX 1 packet. The mixture was added to 50 mL of distilled water and shaken for 3 min and left to settle for 1 min. One mL of the supernatant solution was transferred to an Eppendorf tube and centrifuged at 2000 rpm for 1 min. One hundred μL of the sample extract was mixed with 1500 µL of the sample diluent in a red dilution cup. One hundred µL of diluted sample extract was transferred to the sample cup. The test strip was placed into the sample cup for 6 min for aflatoxins, 9 min for ochratoxin A, 3 min for DON, and 5 min for T2/HT2 and zearalenone, and thereafter it was inserted into the reader according to the instructions of the manufacturer using certain QR codes for each test. 

The limit of detection (LOD) for mycotoxins was given by the company provider at 2 μg/kg for total aflatoxins and ochratoxin A, 0.3 ppb for DON, and 50 ppb for T2/T2H. The results were compared to the maximum limits permitted by the European Commission, which are 5 ppb for ochratoxin (1.2.9, “Unprocessed cereal grains”), 100 ppb for zearalenone (1.5.1, “Unprocessed cereal grains”), and 4 ppb for total aflatoxins (“Aflatoxins” 1.1.12, “Cereals and products derived from cereals”) [13].

### 5.6. Statistical Analysis

Descriptive analysis was employed to evaluate the adherence of labeling information to Palestinian standards. By using SPSS software 24.0 (IBM, Armonk, NY, USA) and one-way ANOVA, the differences in the incidences of mycotoxins, bacterial, and fungal counts in freekeh products were tested. Moreover, Pearson correlations were computed for the incidences of different types of mycotoxins. Duncan’s multiple range test was used to separate the means, and (*p* ≤ 0.05) was considered significant. 

## Figures and Tables

**Figure 1 toxins-16-00499-f001:**
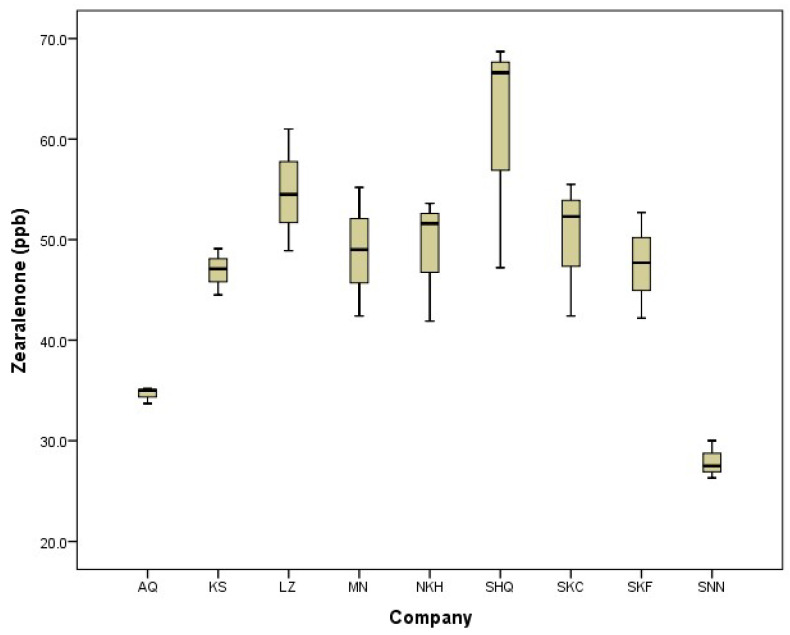
The occurrence of zearalenone (ppb) in freekeh products collected from different local commercial sources.

**Figure 2 toxins-16-00499-f002:**
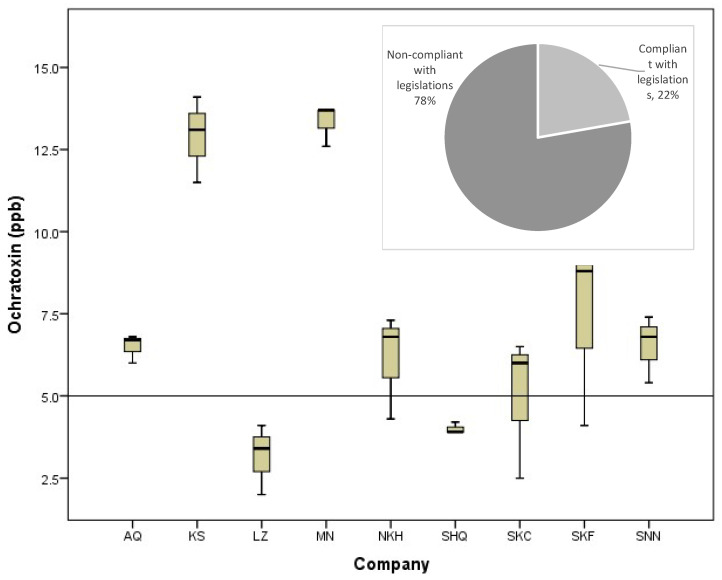
The occurrence of Ochratoxin (ppb) in freekeh products collected from different local commercial sources. The horizontal line is the permitted level according to the EU.

**Figure 3 toxins-16-00499-f003:**
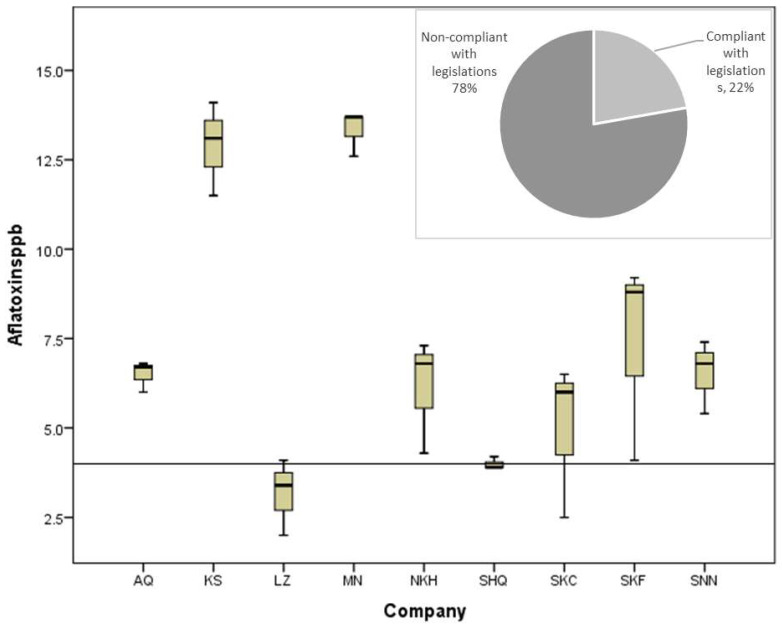
The occurrence of aflatoxins (ppb) in freekeh products collected from different local commercial sources. The horizontal line is the permitted level according to the EU.

**Table 1 toxins-16-00499-t001:** The adherence of freekeh products from different commercial sources to the physical and chemical requirements of the Palestinian Standard for Freekeh (PS 4154/2019).

	Physical Requirements								Chemical Requirements		
Product	Color (Green to Yellow)	Free of Off Odurs	Free of Visible Molds	Free of Weeds	Free of Live Insects	Free of Rodents and Bird Droppings	Impurities (Stones, Soils, Sand) < 0.5%	Impurities (Straw, Foreign Grains) < 2%	Protein Content > 11%	Ash Content < 2.5%	Moisture Content < 13%
AQ	COM *	COM	COM	COM	COM	COM	COM	COM	COM	COM	COM
LZ	COM	COM	COM	COM	COM	COM	COM	COM	COM	COM	COM
KS	COM	COM	COM	COM	COM	COM	COM	COM	COM	COM	COM
MN	COM	COM	COM	COM	COM	COM	COM	NOT **	COM	COM	COM
NKH	COM	COM	COM	COM	COM	COM	COM	COM	COM	COM	COM
SHQ	COM	COM	COM	COM	COM	COM	COM	COM	COM	COM	COM
SKC	COM	COM	COM	COM	COM	COM	COM	COM	NOT	COM	COM
SKF	COM	COM	COM	COM	COM	COM	COM	COM	NOT	COM	COM
SNN	COM	COM	COM	COM	COM	COM	COM	COM	NOT	COM	COM
Overall compliance%	100	100	100	100	100	100	100	89	67	100	100

* Compliant with Palestinian specifications. ** Not compliant with Palestinian specifications.

**Table 2 toxins-16-00499-t002:** The occurrence of different mycotoxins (Zearalenone, DON, T2HT2, aflatoxins, and ochratoxins) in freekeh products collected from different local commercial sources.

Company	Zearalenone (ppb) Mean ± SEM	DON (ppb) Mean ± SEM	T2/HT2 (ppb) Mean ± SEM	Aflatoxins (ppb) Mean ± SEM	Ochratoxin (ppb) Mean ± SEM
AQ	34.63 ^c^ ± 0.47	<0.30 ± 0.00	<50 ± 0.00	6.5 ^bc^ ± 0.25	7.53 ^abc^ ± 0.12
LZ	46.90 ^b^ ± 1.33	<0.30 ± 0.00	<50 ± 0.00	12.90 ^a^ ± 0.75	7.43 ^abc^ ± 0.48
KS	54.80 ^ab^ ± 3.49	<0.30 ± 0.00	<50 ± 0.00	3.17 ^d^ ± 0.61	8.00 ^ab^ ± 0.26
MN	48.87 ^b^ ± 3.67	<0.30 ± 0.00	<50 ± 0.00	13.33 ^a^ ± 0.37	4.63 ^d^ ± 0.46
MKH	49.03 ^b^ ± 3.61	<0.30 ± 0.00	<50 ± 0.00	6.13 ^bc^ ± 0.92	7.17 ^abc^ ± 0.41
SHQ	60.83 ^a^ ± 6.84	<0.30 ± 0.00	<50 ± 0.00	4.00 ^cd^ ± 0.10	6.57 ^bc^ ± 0.33
SKC	50.07 ^ab^ ± 3.94	<0.30 ± 0.00	<50 ± 0.00	5.00 ^bcd^ ± 1.25	8.17 ^a^ ± 0.20
SKF	47.53 ^b^ ± 3.03	<0.30 ± 0.00	<50 ± 0.00	7.36 ^b^ ± 1.63	6.50 ^c^ ± 0.62
SNN	27.93 ^c^ ± 1.08	<0.30 ± 0.00	<50 ± 0.00	6.53 ^bc^ ± 0.59	5.07 ^d^ ± 0.72

Different superscript letters in the same column are different statistically (*p* < 0.05). SEM: standard error of mean.

**Table 3 toxins-16-00499-t003:** Correlation between the occurrence of different mycotoxins (zearalenone, DON, T2HT2, aflatoxins, and ochratoxins) in freekeh products collected from different local commercial sources.

	Zearalenone	Don	T2H2	Ochratoxin	Aflatoxin
Zearalenone	1	a	a	0.26	−0.18
Don		1	a	a	a
T2H2			1	a	a
Ochratoxin				1	−0.45 *
Aflatoxin					1

* Correlation is significant at the 0.05 level. The symbol a indicates that value could not be computed because at least one of the variables was constant.

**Table 4 toxins-16-00499-t004:** Total aerobic bacterial and fungal counts of freekeh products collected from different local commercial sources.

Company	Total Aerobic Count (Log CFU) Mean ± SEM ^1^	Total Fungi (Log CFU ^2^) Mean ± SEM
AQ	3.31 ^ab^ ± 0.36	2.80 ^bc^ ± 0.09
LZ	2.24 ^cd^ ± 0.05	2.66 ^bc^ ± 0.16
KS	1.69 ^de^ ± 0.65	2.10 ^cd^ ± 0.35
MN	1.00 ^e^ ± 0.00	1.39 ^d^ ± 0.55
NKH	4.02 ^a^ ± 0.16	3.29 ^b^ ± 0.27
SHQ	2.13 ^cd^ ± 0.13	2.49 ^bc^ ± 0.03
SKC	2.33 ^cd^ ± 0.12	4.30 ^a^ ± 0.65
SKF	1.85 ^cde^ ± 0.21	2.11 ^cd^ ± 0.16
SNN	2.60 ^bc^ ± 0.22	3.17 ^b^ ± 0.07
*p*-value	<0.05	<0.05

^a–e^ Different superscript letters in the same column are different statistically (*p* < 0.05). ^1^ SEM: standard error of mean; ^2^ CFU: colony forming unit.

## Data Availability

The original contributions presented in this study are included in the article, and further inquiries can be directed to the corresponding author.

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
