# Peer review of "Mycotoxin Prevalence and Microbiological Characteristics of Locally Produced Elected Freekeh Products"

_toxins, 2024, doi:10.3390/toxins16110499_

Round 1

Reviewer 1 Report

Comments and Suggestions for Authors

1.      The paper titled “Mycotoxin prevalence and microbiological characteristics of locally produced freekeh” discussed the occurrence of different types of mycotoxins, physical impurities, and microbiological contamination in Freekeh. The study has potential but need following changes if considered for publication.

2.      Title is good but replace “freekeh” with” elected freekeh products”

3.      Abstract is written poor as it lacks symmetry as methods are not added and results are discussed with less clarity so add methodology and elaborate results along with study recommendations

4.      In abstract, “DON and T2HT2” should be written in full before abbreviations

  1. Introduction is written very short so expand it by adding at least two paragraphs relevant to mycotoxin in freekeh and literature review
  2. Methodology section needs more clarity as proximate analysis are not in detail so add them. Also, microbial analysis is a broader term so divide each category in separate heading with complete elaboration
  3. Results are described in detail but need to be supported with clear understanding of significance and non-significance.
  4. Also, elaborate potential source for contamination of freekeh product and expected control measures for the freekeh
  5. Conclusion need to be more specific with results and need to be elaborated with concrete recommendations for industrial application
  6.  
Comments on the Quality of English Language
  • Grammatical mistakes observed on several places so there is need to go through the paper for language and grammatical mistakes check
  •  

Author Response

I would like to thank your efforts in improving our manuscript.

All comments and suggested corrections by the reviewer have been addressed and corrected accordingly. The following point-to-point our responses to each of the comments. In addition, the changes were highlighted in red color throughout the manuscript. The Quality of English Language has been improved and the changes were highlighted in red color

Reviewer 1

  1. The paper titled “Mycotoxin prevalence and microbiological characteristics of locally produced freekeh” discussed the occurrence of different types of mycotoxins, physical impurities, and microbiological contamination in Freekeh. The study has potential but need following changes if considered for publication.
  2. Title is good but replace “freekeh” with” elected freekeh products”

“elected freekeh products” has been added to the title and it highlighted in red color

  1. Abstract is written poor as it lacks symmetry as methods are not added and results are discussed with less clarity so add methodology and elaborate results along with study recommendations

The abstract has been improved. The methodology was added. Some phrases were restructured to improve clarity. Recommendations were added.

  1. In abstract, “DON and T2HT2” should be written in full before abbreviations

The full names of mentioned mycotoxins were added to the abstract

  1. Introduction is written very short so expand it by adding at least two paragraphs relevant to mycotoxin in freekeh and literature review

The introduction has been extended. Unfortunately, there is lack of studies relevant to mycotoxins in freekeh (I did not get anyone). Huge number of studies are available about wheat grains and wheat flour derived products.

  1. Methodology section needs more clarity as proximate analysis are not in detail so add them. Also, microbial analysis is a broader term so divide each category in separate heading with complete elaboration

Proximate analysis was explained in detail. Moreover, microbiological analysis was split into two sub-section and more details were added. All changes were highlighted in red color

  1. Results are described in detail but need to be supported with clear understanding of significance and non-significance.

The results were described considering significance and non-significance. All changes were highlighted in red color

  1. Also, elaborate potential source for contamination of freekeh product and expected control measures for the freekeh

Potential source for contamination of freekeh product and expected control measures for the freekeh were added

  1. Conclusion need to be more specific with results and need to be elaborated with concrete recommendations for industrial application

Conclusion was modified be more specific with results and recommendations for industrial application

Comments on the Quality of English Language

  • Grammatical mistakes observed on several places so there is need to go through the paper for language and grammatical mistakes check

The Quality of English Language has been improved and the changes were highlighted in red color

Reviewer 2 Report

Comments and Suggestions for Authors

The authors present a manuscript which aims to assess mycotoxin levels in Freekeh. This work fits within the scope of the journal and could be considered for publication but I have some comments regarding the current work.

Abstract - The section on the microbial contamination is not clear to me i.e. fungal and bacterial load. Maybe this should be prior to mycotoxin production in the abstract?

Could give specific information about how the thresholds have been exceeded "In conclusion, a part of locally produced freekeh products were contaminated with mycotoxins and physical impurities at levels that exceeded the local and international permitted limits."

Also could give some indication of the significance of this work and planned future work at end of the abstract

Introduction - At the end of the Introduction could be more specific about the aims and objectives of this study

Could cover previous studies that detail mycotoxins in Freekeh or related grains?

Should introduce the EU levels and explain how they were derived for each mycotoxin in this study.

Results - Table 1 is a bit messy in the labelling of the categories, making it more difficult to interpret

Is it appropriate to detail measurements from companies which are then coded? I'm just wondering if the companies would refute the measurements? Or are the codes non-traceable in which case why not label them company 1, 2, 3 etc.

Also are these levels static on the products or could they be influenced by poor storage etc.

In table 3 you have added a 1 for correllation between similar groups but only for Aflatoxin

It is not clear what the superscripts denote in Table 4. Key does not give enough information

Discussion - 

This part of the discussion is not that clear - line 196 ". In this context, a previous study showed that the incidence level of zearalenone ranged from 51 to 1135 ppb in tested wheat samples in Romania [21]. The highest incidence of zearalenone in wheat samples was 1000 ppb observed by Alexa et al. [31].

The authors compare to a wide range of studies albeit not on the same grains. But an overview of some of these studies would be useful for the introduction.

The discussion doesn't really deal with the limnitations of the current study and the required future work.

Conclusion - needs more information about the exact findings

Methods - 243 typo " from each batch tws samples were collected"

Could be more specific about storage conditions. "The samples were kept at room temperature under dry conditions for further analysis. " Also what is the strogae time and is this consistent forall the products measured. Any correlation between storgae and levels? 

Typo in figure 1 labels

More information? "and placed into the stomacher for 1 min to ensure homogenous distribution"

Could have sub-sections for the methods which relate to different mycotoxins

Comments on the Quality of English Language

There are places where very minor issues do detract slightly from the readability/flow

Author Response

Dear Reviewer

I would like to thank you for your efforts in improving our manuscript.

All comments and suggested corrections by the reviewer have been addressed and corrected accordingly. The following point-to-point our responses to each of the comments. In addition, the changes were highlighted in red color throughout the manuscript. The Quality of English Language has been improved and the changes were highlighted in red color

Comments and Suggestions for Authors

The authors present a manuscript which aims to assess mycotoxin levels in Freekeh. This work fits within the scope of the journal and could be considered for publication but I have some comments regarding the current work.

Abstract - The section on the microbial contamination is not clear to me i.e. fungal and bacterial load. Maybe this should be prior to mycotoxin production in the abstract?

The abstract has been reorganized to impart more clarity.

Could give specific information about how the thresholds have been exceeded "In conclusion, a part of locally produced freekeh products were contaminated with mycotoxins and physical impurities at levels that exceeded the local and international permitted limits."

Specific information related to thresholds of physical impurities as well as mycotoxins was added to the abstract. Moreover, the abstract has been reorganized to impart more clarity.

Also could give some indication of the significance of this work and planned future work at end of the abstract

Firik, also known as freekeh has also has great economic value in the local market, and it has become more viable in the last decade. Accordingly, understanding the current quality, safety, and labeling aspects of locally produced freekeh products can contribute to improving these valuable products.

This is the first study that investigates the occurrence of mycotoxins in locally sourced freekeh products in Palestine. Moreover, the study focuses on evaluating food label compliance with local Palestinian Standard for freekeh (PS 4154/2019) by measuring chemical and physical properties.

The abstract has been modified to stress the significance of this work and planned future work. The changes were highlighted in red color

Introduction - At the end of the Introduction could be more specific about the aims and objectives of this study

The aims and the objectives of the study were modified to be more specific and clearer. The changes were highlighted in red color

Could cover previous studies that detail mycotoxins in Freekeh or related grains?

Unfortunately, there is lack of studies relevant to mycotoxins in freekeh (I did not get anyone). Huge number of studies are available about wheat grains and wheat flour derived products. Some of these studies were added to the introduction.

Should introduce the EU levels and explain how they were derived for each mycotoxin in this study.

EU levels regarding mycotoxins were written in Materials and Methods section with all details. The changes were highlighted in red color

Results - Table 1 is a bit messy in the labelling of the categories, making it more difficult to interpret

Is it appropriate to detail measurements from companies which are then coded? I'm just wondering if the companies would refute the measurements? Or are the codes non-traceable in which case why not label them company 1, 2, 3 etc.

Table 1 contains only descriptive results about the adherence of freekeh products from different commercial sources to the physical and chemical requirements of the Palestinian Standard for Freekeh (PS 4154/2019). The codes do not make any traceability.

Also are these levels static on the products or could they be influenced by poor storage etc.

These levels are almost static and are not highly influenced by poor storage

In table 3 you have added a 1 for correlation between similar groups but only for Aflatoxin

One has been added in the correlations between similar groups

It is not clear what the superscripts denote in Table 4. Key does not give enough information

You are right. Superscripts letters and numbers were added and indicated in footnote of Table 4. The changes were highlighted in red color

Discussion - 

This part of the discussion is not that clear - line 196 ". In this context, a previous study showed that the incidence level of zearalenone ranged from 51 to 1135 ppb in tested wheat samples in Romania [21]. The highest incidence of zearalenone in wheat samples was 1000 ppb observed by Alexa et al. [31].

The authors compare to a wide range of studies albeit not on the same grains. But an overview of some of these studies would be useful for the introduction.

The discussion doesn't really deal with the limitations of the current study and the required future work.

You are right. This is due to the lack of studies about the occurrence of mycotoxins in freekeh products, the results of this study were compared to wheat grains and wheat derived-products. Freekeh is a type of wheat-derived products. Although the topic is not very accurate but it can add some kind of approach.

Some changes were carried out on the text and they were highlighted in red color

Conclusion - needs more information about the exact findings

Conclusion was modified. More information was added as well as the recommendations

Methods - 243 typo " from each batch tws samples were collected"

The typing error was corrected

Could be more specific about storage conditions. "The samples were kept at room temperature under dry conditions for further analysis. " Also, what is the storage time and is this consistent for all the products measured. Any correlation between storage and levels? 

The text was modified. All samples after proper labelling were analyzed on the next day of arrivals. Accordingly, there was no impact of storage conditions on the levels of tested quality traits.

Typo in figure 1 labels

Title of Figure 1 was modified

More information? "and placed into the stomacher for 1 min to ensure homogenous distribution"

More information was added to this section and the changes were highlighted in red color

Could have sub-sections for the methods which relate to different mycotoxins

It is difficult to separate them into sub-sections. Because there are many common steps in the preparation procedures between different types of mycotoxins.

Comments on the Quality of English Language

There are places where very minor issues do detract slightly from the readability/flow

The Quality of English Language has been improved and the changes were highlighted in red color

Reviewer 3 Report

Comments and Suggestions for Authors

Regarding the maximum values allowed by the European Commission, you have reported the values of wheat grains for ochratoxin, zearalenone and Aflatoxin B1,but  only the maximum value allowed for Ochratoxins is correct, of 5 μg/kg , for zearalenone in wheat grains, the maximum value allowed is 100 μg/kg and not 50 μg/kg, this maximum value is for bread, pastries, biscuits, cereal snacks and breakfast cereals. The same applies to total aflatoxins, for wheat grains the maximum permitted value is 4 μg/kg  and not 12.5 μg/kg.  (COMMISSION REGULATION (EU) 2023/915 of 25 April 2023 on maximum levels for certain contaminants in food and repealing Regulation (EC) No 1881/2006 ).

In this case several of the samples analyzed practically exceed these maximum values which are practically lower than the samples marked by you in figure 4 (aflatoxins). In the case of zearalenone the maximum admitted value by EU is 100 ppb (Figure 2). You should re-do Figure 2 and 4  with the maximum admitted values for wheat grains. 

In Materials and Methods,  5.5 Determination of mycotoxins,  the sample preparation is described but not the method of analysis, lateral flow competitive immunochromatographic assay.

Comments on the Quality of English Language

Minor English language requirements

Author Response

Dear reviewer

I would like to thank you for your efforts in improving our manuscript.

All comments and suggested corrections by the reviewer have been addressed and corrected accordingly. The following point-to-point our responses to each of the comments. In addition, the changes were highlighted in red color throughout the manuscript. The Quality of English Language has been improved and the changes were highlighted in red color

Comments and Suggestions for Authors

Regarding the maximum values allowed by the European Commission, you have reported the values of wheat grains for ochratoxin, zearalenone and Aflatoxin B1,but  only the maximum value allowed for Ochratoxins is correct, of 5 μg/kg , for zearalenone in wheat grains, the maximum value allowed is 100 μg/kg and not 50 μg/kg, this maximum value is for bread, pastries, biscuits, cereal snacks and breakfast cereals. The same applies to total aflatoxins, for wheat grains the maximum permitted value is 4 μg/kg  and not 12.5 μg/kg.  (COMMISSION REGULATION (EU) 2023/915 of 25 April 2023 on maximum levels for certain contaminants in food and repealing Regulation (EC) No 1881/2006 ).

In this case several of the samples analyzed practically exceed these maximum values which are practically lower than the samples marked by you in figure 4 (aflatoxins). In the case of zearalenone the maximum admitted value by EU is 100 ppb (Figure 2). You should re-do Figure 2 and 4 with the maximum admitted values for wheat grains. 

You are right. There was misunderstanding for permitted limits. The permitted limits for zearalenone and total aflatoxins were corrected to 100 and 4 ppb, respectively. Based on these changes, the results were also corrected in the text (non-compliance rate) as well as Figure 2 and Figure 4 were corrected. Moreover, the reference has been added

In Materials and Methods, 5.5 Determination of mycotoxins, the sample preparation is described but not the method of analysis, lateral flow competitive immunochromatographic assay.

Materials and Methods have been improved. The text has been modified and highlighted in red color

Comments on the Quality of English Language

Minor English language requirements

The Quality of English Language has been improved and the changes were highlighted in red color

Round 2

Reviewer 1 Report

Comments and Suggestions for Authors

The suggested changes by me are incorporated and paper may be considered for publication

Comments on the Quality of English Language

The language of the paper is appropriate 

Reviewer 2 Report

Comments and Suggestions for Authors

The authors have taken account of my original comments and made significant changes. I believe that the manuscript is improved sufficiently that it could be considered for publication in the current form.

Reviewer 3 Report

Comments and Suggestions for Authors

I have no other comments.

Comments on the Quality of English Language

Minor English language required.